# Level of physical activity and other maternal characteristics during the third trimester of pregnancy and its association with birthweight at term in South Ethiopia: A prospective cohort study

**Meseret Legesse[1]<sup></sup>, Jemal Haider Ali[1]<sup></sup>, Md Dilshad Manzar[2], Mohammed Salahuddin[3], Hamid Yimam Hassen[4,5]\***

1 School of Public Health, College of Health Sciences, Addis Ababa University, Addis Ababa, Ethiopia, 2 Department of Nursing, College of Applied Medical Sciences, Majmaah University, Al Majmaah, Saudi Arabia, 3 Department of Bio-Molecular Sciences, Pharmacology Division, University of Mississippi, Oxford, Mississippi, United States of America, 4 Department of Primary and Interdisciplinary Care, College of Medicine and Health Sciences, University of Antwerp, Antwerp, Belgium, 5 Global Health Institute, College of Medicine and Health Sciences, University of Antwerp, Antwerp, Belgium

☯ These authors contributed equally to this work.
* abdulhamidy71@gmail.com

## Abstract

Birthweight continues to be the leading infant health indicator and the main focus of infant health policy. Low birthweight babies are at a higher risk of mortality and morbidity in most low-income countries. However, the physical activity level of pregnant women and its association with low birthweight is not well studied in Ethiopia. To address the above gap, we aimed to examine the maternal physical activity level and other characteristics during the third trimester and its association with birthweight at term in South Ethiopia. A community-based prospective cohort study was conducted among 247 randomly selected women in their third trimester of pregnancy. We measured the physical activity level using the Global Physical Activity Questionnaire, which included the type and level of various categories of activities. Anthropometric measurements of mothers were taken following standard procedures, and birthweight was recorded within 72 hours of delivery. To identify the effect of physical activity level and other maternal characteristics on low birthweight, we performed a multivariable logistic regression analysis. Overall, 111 (47.2%) mothers were engaged in vigorous physical activities during third trimester. The incidence of low birthweight was 21.6% and 9.68% among newborns of mothers who engaged in vigorous and moderate or low physical activity, respectively. The incidence of low birthweight at term was significantly associated with vigorous physical activity [adjusted odds ratio (AOR) = 2.48; 95% confidence interval (CI): 1.01–6.09], prolonged standing [AOR = 3.37; 95% CI: 1.14–9.93], and squatting [AOR = 2.61; 95% CI: 1.04–6.54]) during the third trimester of pregnancy. The vast majority of pregnant women were engaged in vigorous physical activities in their third trimester. Engagement in vigorous physical activity, standing for longer hours, and squatting were the major contributors to low birthweight at term. Hence, focused counseling should be

**Data Availability Statement:** The data underlying the results presented are uploaded as Supporting Information.

**Funding:** ML has received partial financial support from Addis Ababa University for data collection. All funders had no role in the study design, data collection and analysis, decision to publish, or preparation of the manuscript. There was no additional external funding received for this study.

**Competing interests:** The authors have declared that no competing interests exist.

conducted to reduce vigorous physical activity, standing, and squatting during the third trimester among pregnant women.

## Introduction

Low birthweight (LBW) continued to be a significant public health problem, globally, with the highest prevalence in low- and middle-income countries (LMICs). The United Nations Children's Fund reported a global LBW rate of 14.6%, and more than 91% of LBW infants are born in LMICs [1]. It is an important public health concern, which influences the child's health and nutritional status, physical growth, psychosocial development, and survival [2]. It is among the leading predictors of infant mortality within the first month of life [3]. Later in life, LBW is associated with chronic diseases such as high blood pressure, diabetes, coronary heart disease, and renal insufficiency [4–6]. The prevalence of LBW varies across countries and regions, with higher burden in sub-Saharan Africa and Southeast Asia [7, 8]. In Ethiopia, the prevalence of LBW varies across geographical areas, which ranges from 10% to 28% [9–12]. According to the 2016 Ethiopian Demographic and Health Survey (EDHS), 13% of babies in Ethiopia had LBW [13]. In the same breath, a recent meta-analysis in the country also documented a pooled LBW prevalence of 17.3% [14].

The primary causes for LBW are premature birth and intrauterine growth restriction or a combination of both, in addition to maternal attributes, indicating multifactorial etiologies. Socioeconomic factors such as unfavorable socioeconomic conditions, place of residence, age at pregnancy, maternal level of education, and occupation are risk factors for LBW [9, 10, 15–17]. Physical and behavioral characteristics, including low maternal weight at conception, short maternal stature, maternal comorbidities, absence or inadequate prenatal care, unfavorable reproductive history, absence of antenatal care, birth order and interval, multiple pregnancy, and illicit drug use considerably increase the likelihood of LBW [12, 17–20].

Women from poor communities often engage in heavy physical activities at home and in farms, and they usually have unfavorable birth outcomes, such as LBW, among others [21]. According to a Kenyan study, a majority of women spent 78% of their day-time on engaging in physical activities even during pregnancy [22], and presumably, a similar level of activities also prevails in Ethiopia, which has one of the highest prevalence of LBW.

To reduce the prevalence of LBW, Ethiopia has been implementing various intervention strategies under its National Nutrition Program (NNP), and has signed the Scaling Up Nutrition target to reduce undernutrition, focusing on the first 1,000 days [23]. The level of physical activity during pregnancy and its association with birthweight is widely documented in high-income countries; consequently, these countries developed guidelines on the recommended level of physical activity for pregnant women in each trimester of pregnancy. Nonetheless, such studies in Ethiopia are scant; thus, we aimed to examine the type and level of maternal physical activities and other characteristics during the third trimester and their association with birthweight at term to avail evidences for some programmatic initiatives.

## Methods and materials

### Study setting

The study was conducted at the Butajira Health and Demographic Surveillance Site (HDSS) in Butajira woreda, Southern Nations, Nationalities, and Peoples' Region (SNNPR) of Ethiopia. Butajira woreda is located 130 km away from Addis Ababa, the capital city of Ethiopia. The

Butajira HDSS was established in 1986 and covers nine rural and one urban kebeles (lowest administrative unit in Ethiopia) and has an estimated total population of 76,350. This study was conducted from January to June 2017.

## Study design and study participants

We conducted a prospective cohort study among pregnant women in the Butajira HDSS. Pregnant women who were in their third trimester (31 to 34 weeks of gestation) during the study period were enrolled, while those who had diabetes mellitus, hypertension, and previous preterm baby were excluded to avoid confounding effect. After obtaining the data on sociodemographic characteristics, obstetric history, and physical activity level at baseline, mothers who performed vigorous physical activities were considered as the exposed group, and those who performed moderate or low intensity physical activities were considered as the non-exposed group.

## Sample size and sampling procedure

We calculated the sample size using Epi-info version 7 StatCalc sample size calculation package for cohort studies, assuming an estimated LBW prevalence of 9.8% among women in the unexposed group, as reported in SNNPR [13]; a 15% difference between the exposed and unexposed group; an 80% power; and a 95% level of confidence. Adding 10% non-response, the final sample size was determined to be 247 women.

All 10 kebeles in the Butajira HDSS site were selected purposively, considering the better follow-up experience of the population. In order to recruit the participants, we prepared a sampling frame. It constituted a list of names of pregnant women who were thought to be in their third trimester and their detailed information, which included their house number and names of kebeles from the HDSS pregnancy survey list. Using the list, the mothers were visited at home, and based on their last menstrual period (LMP), women in their third trimester, particularly 31 to 34 weeks were identified. The sample size was distributed proportionally based on the number of third trimester mothers living in each kebele. Then, the participants were selected from each cluster using simple random sampling.

## Data collection procedure

Data were collected through face-to-face interview, using a structured and pretested questionnaire, which was adapted from the EDHS and other peer reviewed articles. The height of the mother was measured to the nearest 0.1 cm using a wooden stadiometer with a sliding head bar and the mother on barefoot. Mid-upper arm circumference (MUAC) was also measured to the nearest 0.1 cm on the left arm, using an adult MUAC tape, following standard procedure. Pre-pregnancy weight was self-reported, and current weight was measured using a digital weighing scale. The difference between pre-pregnancy weight and current weight was used for estimating the weight gain during pregnancy. The main outcome, weight of the newborn, was measured by trained data collectors within 72 hours after birth using a digital weighing scale. To assess the level of physical activity, we employed the modified global physical activity questionnaire, standardized for low-income countries [24]. The questionnaire measured the level of physical activity of the women within the last 7 days prior to enrollment and contains a list of activities with different domains, frequencies of each activity per week, and durations per day. The type of physical activity at work, travel, and sports or leisure time activities were assessed. The level was then classified as vigorous activities, moderate activities, walking, and sitting/reclining. Then, we dichotomized women into those who performed vigorous physical activities and those who do not (including moderate and low physical activity level). Positions like standing for longer

hours, squatting during routine daily physical activity, and lifting heavy loads were also assessed. We assessed the dietary habit of the mother during pregnancy using a food frequency questionnaire (FFQ). The FFQ was prepared to measure the estimated frequency of consumption for a specific food type within one month prior to the baseline survey.

## Data quality control measures

We conducted a 2-day training on interview techniques, physical activity, and anthropometric measurements. The questionnaire was translated into the local language and reviewed for consistency. We also conducted a pilot test in an adjacent zone called '*Silte*', which has similar socioeconomic, cultural, and geographical characteristics as those of the study area. We performed an internal consistency test using the pilot data, and results showed acceptable level (Cronbach's alpha = 0.79). The collected data were assessed by field supervisors on a daily basis for incompleteness and/or inconsistencies. The anthropometric and birthweight measurement scales were calibrated frequently.

## Definition of terms

Standing for a long period was defined as standing for more than 3 hours per day on average. Vigorous activities were described as activities that require a large amount of physical effort and largely increase breathing or heart rate (e.g., carrying or lifting heavy loads, digging, and construction work). Meanwhile, moderate activities were described as activities that require moderate physical effort and mildly increase breathing or heart rate (e.g., brisk walking and carrying light loads). Sedentary behavior was defined as sitting or reclining at work or at home; while getting to and from places; travelling in a car, bus, motor, or cart; reading; or watching television.

## Statistical methods

The data, upon checks for completeness and consistency, were entered and cleaned using EpiData version 3.1., and then exported to STATA version 14.0 for further processing and analysis. Of the 247 participants enrolled in this study, 11 (4.5%) did not have information on birthweight and 1 (0.4%) born preterm and thus were removed from the analysis. Descriptive statistics including frequencies with percentage or means and standard deviations (SD) were computed as needed. To determine the wealth index, a principal component analysis of housing infrastructure and ownership of household assets was performed, and the score was consequently divided into five quintiles (lowest, second, middle, fourth, and highest). Depending on their engagement in daily physical activity, mothers were categorized into the vigorous intensity physical activity group and moderate or low intensity physical activity group. A bivariable analysis was carried out to determine the crude association of predictor variables with low birthweight, and based on the results, variables significant at p-value <0.05 were selected for multivariable analysis. To identify the major determinants of LBW, a multivariable logistic regression was performed. The variable antenatal care (ANC) follow up was excluded from the multivariable model due to collinearity with iron folic acid supplementation. The adjusted odds ratio (AOR) with its 95% confidence interval (CI) was used to determine the strength and significance of the association. A p-value of <0.05 was considered statistically significant.

## Participant consent and ethical approval

The protocol of this study was approved by the institutional review board of the College of Health Sciences, Addis Ababa University. Written informed consent and parental assent were

obtained from the participants after they received explanations about the possible risks, benefits, issue of confidentiality, voluntarism, and purposes of the study. The study is in compliance with the principles of the declaration of Helsinki.

## Results

A total of 247 women at their third trimester of pregnancy were recruited for this study, and 235 of them were included in the final analysis, yielding a retention rate of 95.1%. Meanwhile, 111 (47.2%) and 124 (52.8%) mothers in the third trimester of pregnancy were included in the vigorous and moderate/low physical activity groups, respectively.

### Socioeconomic characteristics of the mothers

The socioeconomic characteristics of the mothers are presented in Table 1. The mean age of the respondents was 29.1 years (SD: 5.4), ranging from 17 to 45 years. Most (71.1%) of them

**Table 1. Socioeconomic characteristics of pregnant women and the incidence of low birthweight in Butajira, Ethiopia.**

| Maternal characteristics | Total sample N (%) | Incidence of low birthweight n (%) | p-value [a] |
|---|---|---|---|
| Total | 235 (100) | 36 (15.3) | |
| *Age (y)* | | | |
| 15–24 | 45 (19.1) | 5 (11.1) | 0.02 |
| 25–34 | 140 (59.6) | 17 (12.1) | |
| 35–45 | 50 (21.3) | 14 (28.0) | |
| *Residence* | | | |
| Rural | 167 (71.1) | 31 (18.6) | 0.03 |
| Urban | 68 (28.9) | 5 (7.4) | |
| *Ethnicity* | | | |
| Gurage | 152 (64.7) | 25 (16.4) | 0.69 |
| Silte | 52 (22.1) | 6 (11.5) | |
| Others** | 31 (13.2) | 5 (16.1) | |
| *Formal education* | | | |
| No | 113 (48.1) | 24 (21.2) | 0.015 |
| Yes | 122 (51.9) | 12 (9.8) | |
| *Husbands' formal education* | | | |
| No | 85 (36.2) | 20 (23.5) | 0.009 |
| Yes | 150 (63.8) | 16 (10.7) | |
| *Water source* | | | |
| Improved | 191 (81.3) | 24 (12.6) | 0.015 |
| Non-improved | 44 (18.7) | 12 (27.3) | |
| *Wealth quintile* | | | |
| Lowest | 47 (20.0) | 7 (14.9) | 0.845 |
| Second | 48 (20.4) | 9 (18.6) | |
| Middle | 46 (19.6) | 8 (17.4) | |
| Fourth | 47 (20.0) | 7 (14.9) | |
| Highest | 47 (20.0) | 5 (10.6) | |

*Other = Catholic, Protestant, and other traditional religions

**Other = Oromo, Amhara, Wolayta, and Hadiya

[a]Pearson's chi-square test

were rural residents, and 18.6% gave birth to a LBW baby. Over three-quarters (78.3%) of mothers were Muslim, and 152 (64.7%) belonged to the Gurage ethnic group. Nearly half (48.1%) of the mothers and 85 (36.2%) of their husbands had no formal education. Most (81.3%) of them used an improved drinking water source. The incidence of LBW significantly varied across age groups, with a higher incidence (28.0%) observed among the age group of 35 to 45 years (p = 0.02). Similarly, a significant variation was observed in the place of residence (p = 0.03), mothers' level of education (p = 0.015), husbands' level of education (p = 0.009), and water source (p = 0.015) between groups.

## Mothers' obstetric history, behavioral characteristics, and anthropometric measurement

Table 2 displays the obstetric history, behavioral and anthropometric characteristics of the mother and the incidence of low birthweight. Most (84.3%) of the mothers were multigravida, 130 (65.7%) had three or more children, and 23 (9.8%) had a history of miscarriage in their lifetime. More than two-fifths (42.9%) gave birth within an interval of 23 months or less. The vast majorities (90.6%) of mothers attended at least one ANC follow-up, and 176 (74.9%) took iron and folic acid supplementation. Almost all (97.4%) of the mothers consumed coffee, and

**Table 2. Obstetric and behavioral characteristics of pregnant women in their third trimester in Butajira Ethiopia.**

| Obstetric and behavioral characteristics | Total sample N (%) | Incidence of LBW n (%) | p-value |
|---|---|---|---|
| Total | 235 (100) | 36 (15.3) | |
| **Gravidity [a]** | | | |
| Multigravida | 198 (84.3) | 31 (15.7) | 0.740 |
| Primigravida | 37 (15.7) | 5 (13.5) | |
| **Parity (n = 198)[a]** | | | |
| 1–2 children | 68 (34.3) | 9 (13.2) | 0.498 |
| ≥3 children | 130 (65.7) | 22 (16.9) | |
| **History of miscarriage [a]** | | | |
| Yes | 23 (9.8) | 8 (34.8) | 0.060 |
| No | 212 (90.2) | 28 (13.2) | |
| **Birth interval (months) (n = 196)[a]** | | | |
| <23 | 84 (42.9) | 15 (17.9) | 0.498 |
| ≥24 | 112 (57.1) | 16 (14.3) | |
| **Antenatal care[a]** | | | |
| Yes | 213 (90.6) | 29 (13.6) | 0.024 |
| No | 22 (9.4) | 7 (31.8) | |
| **Iron folic acid supplement [a]** | | | |
| Yes | 176 (74.9) | 18 (10.2) | <0.001 |
| No | 59 (25.1) | 18 (30.5) | |
| **Coffee consumption [b]** | | | |
| Yes | 229 (97.4) | 35 (15.3) | 0.926 |
| No | 6 (2.6) | 1 (16.7) | |
| **Khat[$] chewing [a]** | | | |
| Yes | 79 (33.6) | 13 (16.5) | 0.731 |
| No | 156 (66.4) | 23 (14.7) | |
| **Maternal height (cm) [a]** | | | |

(*Continued*)

**Table 2.** (Continued)

| Obstetric and behavioral characteristics | Total sample N (%) | Incidence of LBW n (%) | p-value |
|---|---|---|---|
| <155 | 43 (18.3) | 11 (25.6) | 0.039 |
| ≥155 | 192 (81.7) | 25 (13.0) | |
| *Maternal mid-upper arm circumference* [a] | | | |
| <23 | 50 (21.3) | 13 (26.0) | 0.018 |
| ≥23 | 185 (78.7) | 23 (12.4) | |
| *Pre-pregnancy weight (n = 61)* [b] | | | |
| <50 kg | 11 (4.7) | 3 (27.3) | 0.035 |
| 50–60 kg | 41 (17.4) | 2 (4.9) | |
| ≥60 kg | 9 (3.8) | 0 (0.0) | |
| *Gestational weight gain* [b] | | | |
| ≤7 kg | 25 (10.6) | 3 (12.0) | 0.1958 |
| 8–12 kg | 20 (8.5) | 0 (0.0) | |
| >12 kg | 5 (2.1) | 1 (25.00) | |

[$] Catha edulis

[a] Pearson's chi-square test

[b] Fisher's exact test

79 (33.6%) consumed khat (*Catha Edulis*). Mothers with a history of miscarriage had a significantly higher incidence of LBW (34.8%) than those who did not have a history of miscarriage (13.2%) ($p < 0.01$). Similarly, the incidence varied depending on the use of ANC ($p < 0.05$) and iron and folic acid supplementation ($p < 0.001$). Nearly one-fifth (18.3%) of the mothers had a height of less than 155 cm, and 50 (21.3%) had a MUAC of less than 23 cm. Eleven (4.7%) mothers weighed less than 50 kg, and 25 (10.6%) gained less than 7 kg during pregnancy. The incidence of LBW varied with maternal height ($p = 0.04$), MUAC ($p = 0.02$), and pre-pregnancy weight ($p = 0.04$).

## Food types and consumption frequency

The food consumption frequency of mothers during pregnancy is summarized in Table 3. Majority (91.1%) and 29 (12.3%) of the mothers consumed cereal and enset (*Ensete ventricosum*)-based food, respectively, at least once a day. More than half (56.6%) and 24 (10.2%) of them consumed vegetables and fruits at least once a day. A total of 106 (45.1%) and 62 (26.4%) of them consumed meat and milk products, respectively, less than once a month. More than two-fifth (41.7%) and 53 (22.6%) of them consumed egg and legumes, respectively, at least once a day. No significant association was observed in the incidence of LBW and consumption of various foods groups.

## Physical activity level and incidence of low birthweight

A total of 111 (47.2%) mothers were engaged in vigorous physical activities during their third trimester. The mean birthweight for newborns of mothers in the vigorous physical activity group was 2,917 grams (SD: 472), while that of newborns of mothers in the moderate or low physical activity group was 3,024 (SD: 376) grams. The incidence of LBW was significantly higher among mothers in the vigorous activity group 24 (21.6%) than moderate or low activity group 12 (9.7%) ($p = 0.011$). A total of 63 (26.8%) and 34 (14.5%) mothers had history of squatting and standing for longer hours, respectively, in their third trimester. The incidence of

**Table 3. Food consumption frequency among pregnant women in their third trimester in Butajira, Ethiopia.**

| Food items | Total sample N (%) | Incidence of LBW n (%) | p-value |
|---|---|---|---|
| **Total** | 235 (100.0) | 36 (15.3) | |
| ***Cereals and bread*** [a] | | | |
| ≥once/day | 214 (91.1) | 30 (14.0) | 0.077 |
| <once/day | 21 (8.9) | 6 (28.6) | |
| ***Enset based*** [a] | | | |
| ≥once/day | 29 (12.3) | 5 (17.2) | 0.8598 |
| 4–6 times/week | 15 (6.4) | 2 (13.3) | |
| 2–3 times in a week | 33 (14.0) | 5 (15.2) | |
| Once in a week | 76 (32.3) | 14 (18.4) | |
| Twice or less in a month | 82 (34.9) | 10 (12.2) | |
| ***Vegetables*** [a] | | | |
| ≥once/day | 133 (56.6) | 19 (14.3) | 0.090 |
| 4–6 times/week | 43 (18.3) | 11 (25.6) | |
| 2–3 times/week | 59 (25.1) | 6 (10.2) | |
| ***Fruits*** [b] | | | |
| ≥once/day | 24 (10.2) | 2 (8.3) | 0.4007 |
| 4–6 times/week | 44 (18.7) | 7 (15.9) | |
| 2–3 times/week | 38 (16.2) | 4 (10.5) | |
| Once a week | 46 (19.6) | 11 (23.9) | |
| Once a month | 31 (13.2) | 3 (9.7) | |
| <1 in a month | 42 (17.9) | 9 (21.4) | |
| ***Meat*** [b] | | | |
| ≥once/day | 13 (5.5) | 2 (15.4) | 0.2743 |
| 4–6 times/week | 28 (11.9) | 2 (7.1) | |
| Once a week | 33 (14.0) | 3 (9.1) | |
| 1–2 times/month | 55 (23.4) | 7 12.7() | |
| <1 in a month | 106 (45.1) | 22 (20.8) | |
| ***Egg*** [a] | | | |
| ≥once/day | 98 (41.7) | 15 (15.3) | 0.9963 |
| 2–3 times/week | 137 (58.3) | 21 (15.3) | |
| ***Legumes*** [b] | | | |
| ≥once/day | 53 (22.6) | 10 (18.9) | 0.054 |
| 4–6 times/week | 4 (1.7) | 2 (50.0) | |
| 2–3 times/week | 70 (29.8) | 5 (7.1) | |
| Once /week | 18 (7.7) | 3 (16.7) | |
| 1–2 times/month | 20 (8.5) | 2 (10.0) | |
| <1 in a month | 51 (21.7) | 14 (27.5) | |
| ***Milk, cheese, and yoghurt*** [a] | | | |
| ≥once/day | 41 (17.4) | 2 (4.9) | 0.1908 |
| 2–3 times/week | 75 (31.9) | 15 (20.0) | |
| 1–2 times/month | 57 (24.3) | 9 (15.8) | |
| <1 in a month | 62 (26.4) | 10 (16.1) | |
| ***Oils and fat*** [a] | | | |
| ≤once/day | 173 (73.6) | 26 (15.0) | 0.7634 |
| 2–3 times/week | 31 (13.2) | 4 (12.9) | |
| Once in a week | 31 (13.2) | 6 (19.4) | |
| ***Sweets*** [a] | | | |

(*Continued*)

**Table 3.** (Continued)

| Food items | Total sample N (%) | Incidence of LBW n (%) | p-value |
|---|---|---|---|
| >once/day | 52 (22.1) | 8 (15.4) | 0.9795 |
| 4–6 times/week | 67 (28.5) | 11 (16.4) | |
| 2–3 times/week | 44 (18.7) | 7 (15.9) | |
| Once in a week | 72 (30.6) | 10 (13.9) | |

a = Pearson's chi-square test

b = Fisher's exact test

LBW was higher in those who stood for longer time (p = 0.013) and squatted (p = 0.009). The incidence of LBW was 25 (21.2%) and 4 (14.3%) in mothers who walked more than 60 minutes and below 30 minutes per day, respectively (p = 0.217). With regard to the level of sedentary behavior and sleeping, 34 (94.4%) and 26 (72.2%) of the mothers who gave birth to LBW babies were those who sat or reclined for <165 minutes per day (15.7%) and slept ≥8 hours (16.4%), respectively.

## Multivariable analysis of physical activity and other determinants of low birthweight

To identify the independent effects of pattern and intensity of physical activity level on LBW, we employed a multivariable logistic regression model. It showed that the incidence of LBW significantly varied according to the level of vigorous physical activity level, standing for longer hours, walking, and squatting during the third trimester. As expected, water source and iron and folic acid supplementation were also significantly associated with LBW. The probability of LBW was 2.5 times higher for mothers who were involved in vigorous physical activities during the third trimester than those involved in moderate or low physical activities [AOR = 2.48, 95% CI: 1.01–6.09]. Similarly, the likelihood of LBW was 3.4 times higher for mothers who were involved in activities that require prolonged standing than those who were not involved in these types of activities [AOR = 3.37, 95% CI: 1.14–9.93]. The risk of LBW was 2.6 times higher in mothers who performed squatting in their third trimester than those who did not perform squatting [AOR = 2.61, 95% CI: 1.04–6.54]. (Table 4)

## Discussion

The sustainable development goal targeted to reduce neonatal mortality to at least 12 per 1,000 live births by 2030. Likewise, the Global Nutrition Target also aimed to achieve a 30% reduction in the number of infants born with a weight lower than 2,500 gm by the year 2025. As part of the LBW reduction strategy, it is indispensable to identify the risk factors linked to it in order to develop a targeted intervention. With this background information, we conducted a prospective cohort study among pregnant women engaged in physical activity during their third trimester, to determine the magnitude and its attributors on the incidence of LBW.

Based on our study, the incidence of LBW was 15.32%, which is consistent with the 2016 EDHS national estimate [13]. Moreover, other related studies in the country reported a LBW incidence of 13%–16.5% [10, 25, 26]. Our estimate is lower than that obtained from previous studies performed in countries with a documented prevalence of 22–54% [27–29]. Such discrepancies were expected since most of the above studies were facility based and targeted the high-risk groups. Compared with some previous studies conducted in Gondar and Jimma cities [30, 31], the values observed in the present study were higher due to the variations across

**Table 4. Bivariable and multivariable analysis of the association of physical activity, sociodemographic, and reproductive health characteristics with low birth weight in South Ethiopia.**

| Variable | Low birthweight | | COR [95% CI] | AOR [95% CI] |
|---|---|---|---|---|
| | Yes n (%) | No n (%) | | |
| *Age group (y)* | | | | |
| 15–24 | 5 (11.1) | 40 (88.9) | 0.90 [0.31–2.61] | 1.08 [0.33–3.51] |
| 25–34 | 17 (12.1) | 123 (87.9) | 1 | 1 |
| 35–45 | 14 (28.0) | 36 (72.0) | 2.81 [1.26–9.49] | 2.36 [0.82–6.75] |
| *Residence* | | | | |
| Rural | 31 (18.6) | 136 (81.4) | 2.87 [1.06–7.73] | 2.07 [0.60–7.05] |
| Urban | 5 (7.4) | 63 (92.6) | 1 | 1 |
| *Educational status* | | | | |
| Illiterate | 24 (21.2) | 89 (78.8) | 2.47 [1.17–5.22] | 0.92 [0.32–2.61] |
| Literate | 12 (9.8) | 110 (90.2) | 1 | 1 |
| *Husband's educational status* | | | | |
| Illiterate | 20 (23.5) | 65 (76.5) | 2.58 [1.25–5.30] | 1.98 [0.92–4.99] |
| Literate | 16 (10.7) | 134 (89.3) | 1 | 1 |
| *Water source* | | | | |
| Improved | 24 (12.6) | 167 (87.4) | 1 | 1 |
| Unimproved | 12 (27.3) | 32 (72.7) | 2.60 [1.18–5.74] | 3.16 [1.17–8.52]* |
| *Iron folic acid supplementation* | | | | |
| Yes | 18 (10.2) | 158 (89.8) | 1 | 1 |
| No | 18(30.5) | 41 (69.5) | 3.85 [1.84–8.06] | 6.13 [2.53–14.84]* |
| *Mid-upper arm circumference* | | | | |
| <23 | 13 (26.0) | 37 (74.0) | 2.47 [1.15–5.33] | 2.01 [0.87–6.21] |
| ≥23 | 23 (12.4) | 162 (87.6) | 1 | 1 |
| *Physical activity level* | | | | |
| Vigorous activity | 24 (21.6) | 87 (78.4) | 2.57 [1.21–5.43] | 2.48 [1.01–6.09]* |
| Moderate activity | 12 (9.7) | 112 (90.3) | 0.10 [0.05–0.19] | 1 |
| *Standing for longer hours* | | | | |
| Yes | 10 (29.4) | 24 (70.6) | 2.80 [1.20–6.52] | 3.37 [1.14–9.93]* |
| No | 26 (12.9) | 175 (87.1) | 1 | 1 |
| *Squatting* | | | | |
| Yes | 16 (25.4) | 47 (74.6) | 2.58 [1.24–5.39] | 2.61 [1.04–6.54]* |
| No | 20 (11.6) | 152 (88.4) | 1 | 1 |

*Statistically significant association at $p < 0.05$

geographical settings and seasonality. A recent systematic review performed in Ethiopia documented a pooled estimate of 17.3%, which is slightly higher than our findings [14], suggesting that LBW remains an important public health problem in the country.

In our study, 47.2% of mothers were involved in vigorous physical activities at work or home, and a significant proportion of them took part in activities that required squatting and standing for longer hours during their third trimester. Moreover, being involved in high intensity physical activity along with the warm weather would consequently lead to small for gestational age fetus and low birthweight [32].

In the multivariable analysis, engaging in vigorous intensity physical activities, prolonged standing, squatting during the third trimester, using unimproved water sources, and not taking iron supplements during pregnancy were significant determinants of LBW at term. The

incidence of LBW was higher among mothers involved in vigorous activities, and our finding is concordant with McCowan et al.'s findings who documented that daily vigorous physical activity is associated with higher risk of LBW [33]. Bisson et al. also reported a 19.8 gram reduction in birthweight for one metabolic equivalent of task (MET)/hours/week increment spent in any vigorous exercise [32]. Moreover, Magann and his colleagues showed that women who engaged in heavy exercise had smaller (86.5 g) infants than sedentary women, which again emphasizes how vigorous activities affect birthweight [34]. By contrast, few studies indicated no significant effect of vigorous physical activity on birthweight [35–37]. These findings need to be interpreted cautiously since these studies assessed the effects of planned physical activity alone, which was 2 to 3 times per week for not more than 1 hour.

Mothers involved in activities that require prolonged standing were also more likely to have LBW neonates, and this finding is in congruence with the systematic review report, which showed that uninterrupted standing increased the odds of LBW threefold [38]. Increase in the activity of the sympathetic nervous system in active muscles, following prolonged standing results in the return of blood from visceral arteries to active muscles, increased sweating, decreased plasma volume, and reduced perfusion of blood to uterine and placental arteries [39].

The odds of having LBW was higher among mothers who performed squatting and is again in line with the findings of the Thailand and Indian studies, which showed that women who performed activities in squatting position were more likely to give birth to LBW babies [40, 41].

Using water from non-improved source was found to increase the odds of LBW babies, and similar finding was reported by a study from Ghana, which documented that living in a community with low coverage of safe water supply is associated with a high prevalence of LBW [42]. It is likely that mothers who do not have improved water supply at their locality might travel far to bring water. This causes physical strain that could result in low gestational weight gain, a known risk factor for LBW birth.

Mothers who did not receive iron supplements during pregnancy were more likely to have LBW babies. Our result is consistent with some previous studies conducted in Ethiopia [10, 43] and Nepal [44] that showed that by supplementing mothers with iron and other micronutrients, the risk of LBW could be reduced. This is sufficient evidence to support the fact that iron folic acid supplementation is associated with normal birthweight [45].

The following limitations need to be considered when interpreting the findings of this study. First, the use of self-reported physical activity could lead to subjective memory and reporting variations. However, as the variation is non-differential, i.e., it is not dependent on the outcome, the effect size estimate will not be biased. Second, we did not estimate the MET; hence, we could not measure the amount of reduction in birthweight for every unit increase in MET. Third, due to practical difficulty, we did not measure the exact amount of food servings consumed. However, the relative food frequency indicates minimal or no variation in food intake between study groups. Fourth, due to low educational status, only 25% of women reported pre-pregnancy weight. Thus, we could not adjust for pre-pregnancy BMI. Finally, we did not estimate the pregnancy weight gain at each trimester of pregnancy, which might have an impact on birthweight. However, as pregnancy weight gain might be in the causal pathway, controlling for it could lead to a biased estimate. Therefore, we believe that the effect size for vigorous physical activity is still valid.

## Conclusion

The incidence of LBW in Ethiopia is considerably high. A significant proportion of women continue engaging in activities requiring vigorous physical effort, prolonged standing and

squatting during the span of their pregnancy. These activities were found to be significantly associated with LBW in their babies. Lack of access to an improved water source and poor iron folic acid supplement utilization were linked with LBW. Therefore, health professionals working in maternity units should perform counseling regarding the recommended level of physical activity in each trimester of pregnancy. Moreover, the National Nutrition Program needs to improve the optimal dose of iron folic acid supplementation for pregnant women. Further research with a larger sample size and objective measurement of physical activity in each trimester is recommended.

## Supporting information

**S1 Table. STROBE statement—checklist of items for cohort studies.**
(DOCX)

**S1 File. The English and Amharic version of the questionnaire used in this study.**
(PDF)

**S1 Dataset. The minimal dataset of the study.**
(ZIP)

## Acknowledgments

The authors would like to thank the Butajira Health and Demographic Surveillance Site for their cooperation. Moreover, we would like to thank village informants for their cooperation in identifying pregnant women. We are also grateful to all study participants for their willingness to participate in the study.

## Author Contributions

**Conceptualization:** Meseret Legesse, Jemal Haider Ali.

**Data curation:** Meseret Legesse, Jemal Haider Ali, Md Dilshad Manzar, Hamid Yimam Hassen.

**Formal analysis:** Meseret Legesse, Hamid Yimam Hassen.

**Funding acquisition:** Meseret Legesse.

**Investigation:** Meseret Legesse, Jemal Haider Ali, Hamid Yimam Hassen.

**Methodology:** Meseret Legesse, Jemal Haider Ali, Md Dilshad Manzar, Mohammed Salahuddin, Hamid Yimam Hassen.

**Project administration:** Meseret Legesse, Jemal Haider Ali.

**Resources:** Meseret Legesse, Jemal Haider Ali, Mohammed Salahuddin.

**Software:** Meseret Legesse, Hamid Yimam Hassen.

**Supervision:** Jemal Haider Ali, Hamid Yimam Hassen.

**Validation:** Meseret Legesse, Jemal Haider Ali, Mohammed Salahuddin, Hamid Yimam Hassen.

**Visualization:** Meseret Legesse, Jemal Haider Ali, Md Dilshad Manzar, Hamid Yimam Hassen.

**Writing – original draft:** Meseret Legesse, Hamid Yimam Hassen.

**Writing – review & editing:** Jemal Haider Ali, Md Dilshad Manzar, Mohammed Salahuddin, Hamid Yimam Hassen.

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
