## [Decision Letter · Decision Letter 0]

18 May 2020

PONE-D-20-09704

Level of physical activity during the third trimester of pregnancy and its association with birthweight at term in South Ethiopia: A prospective cohort study

PLOS ONE

Dear Mr. Hassen,

Thank you for submitting your manuscript to PLOS ONE. After careful consideration, we feel that it has merit but does not fully meet PLOS ONE’s publication criteria as it currently stands. Therefore, we invite you to submit a revised version of the manuscript that addresses the points raised during the review process.

SPECIFIC ACADEMIC EDITOR COMMENTS: Thank you for submitting your manuscript. Two expert reviewers handled your manuscript. Although interest was found in your study, there were major comments that arose during review. A number of these comments relate to the need for clarification and expansion of several vague points throughout the manuscript.

We would appreciate receiving your revised manuscript by Jul 02 2020 11:59PM. To enhance the reproducibility of your results, we recommend that if applicable you deposit your laboratory protocols in protocols.io, where a protocol can be assigned its own identifier (DOI) such that it can be cited independently in the future. For instructions see: http://journals.plos.org/plosone/s/submission-guidelines#loc-laboratory-protocols

We look forward to receiving your revised manuscript.

Kind regards,

Frank T. Spradley

Academic Editor

PLOS ONE

ML has received partial financial support from Addis Ababa University for data

collection. The funder had no role in study design, data collection and analysis,

decision to publish, or preparation of the manuscript.

5. Please include a copy of Table 2 which you refer to in your text on page 11.

Reviewers' comments:

Reviewer's Responses to Questions

**Comments to the Author**

1. Is the manuscript technically sound, and do the data support the conclusions?

Reviewer #1: Partly

Reviewer #2: Yes

2. Has the statistical analysis been performed appropriately and rigorously? 

Reviewer #1: I Don't Know

Reviewer #2: Yes

3. Have the authors made all data underlying the findings in their manuscript fully available?

Reviewer #1: Yes

Reviewer #2: Yes

4. Is the manuscript presented in an intelligible fashion and written in standard English?

Reviewer #1: Yes

Reviewer #2: Yes

5. Review Comments to the Author

Reviewer #1: Thank you for the opportunity to review this paper. The paper examines the relationship between physical activity and low birth weight in a cohort of women in Ethiopia. The topic is important as LBW is a significant contributor to newborn mortality and morbidity and had longer impacts on childhood growth and development.

More detail is needed in a number of areas.

What gestation were women recruited? The third trimester is too broad and better clarity is needed. How was the gestation determined and how sure are the authors of the accuracy of the gestation?

The inclusion states that women with a preterm baby and multiple birth were excluded. How were multiple births determined? Does the preterm birth comment mean women with a history of previous preterm birth were excluded as whether they will have a preterm birth cannot be known at this point?

Clarity about when the physical activity was taken from is needed. Many women do slow down towards the end of pregnancy so their activity level in early pregnancy may be different to late pregnancy. These things are not statistic necessarily. Was this accounted for at all?

In the analysis section, a clearer explanation of how the variable for the multivariate analysis were selected is needed. What was the cut off point? Which ones were initially included and then removed?

Parity does not seem to have had much attention. This is important as it is associated with a number of important issues including anaemia. Why was it not included in the multivariate analysis?

Abortion is included which is important. Is this spontaneous abortion which would be better mentioned as a miscarriage or an induced abortion? The long term implications of these two are different and so need to be clarified.

BMI is a common measure in studies like this but I cannot see this included.

The food consumption analysis is comprehensive but not easy to interpret. There are also too many individual comparisons related to fruit and vegetables etc consumed. Can these data be summarized into adequate diet versus not adequate diet?

There are more than 30 individual comparisons which makes the chance of finding an error when one does not really exist high, especially given the sample size. Can the authors comment on the power of the study given the number of comparisons and the possibly of statistical error?

The multivariate analysis seems to be missing important issues – BMI in early pregnancy, parity, anaemia? Were interactions examined? It is possible that there is an interaction between physical activity, standing for long hours and squatting. Given these are the three outcomes with significant results further examination needs to occur.

Some of the language needs attention. For example, scant rather than scanty (line 76), women instead of patients (line 99) and gave birth to rather than delivered (line 180).

Reviewer #2: Comments to the authors

The manuscript reports a prospective cohort study addressing the association of the level of physical activity with birthweigh in Ethiopia, where there is a high incidence of low birthweigh, and women work and house conditions usually lead to high volumes and intensities of physical activity. The study is well designed, the methodology is well explained, and the paper is well written and with a good level of English. Every section contains the necessary information and they are well related to each other. Just a few comments and small corrections have been suggested:

OVERVIEW:

The study seems quite complete. However, even if physical activity is the main outcome, I believe the study covers much more than physical activity. The study provides additional information about how other factors may be associated with low birthweigh, such as socioeconomic, obstetric and behavioral characteristics, food consumption and iron and folic acid supplementation. Therefore, I would extend the title of the paper to show that other factors have been studied, for instance: “Level of physical activity and other maternal characteristics during the third trimester of pregnancy and its association with birthweight at term in South Ethiopia: A prospective cohort study”. If the suggestion is followed, I would also mention this in the abstract.

METHODS

- Please, explain when this study was carried out.

- Please, explain if the modified version of the global physical activity questionnaire was validated or published somewhere. Was the pilot test in Silte published?

- Does the questionnaire consider the intensity of the physical activity developed? Or was the intensity deduced according to the type of exercise? “The level of physical activity was classified as moderate activity, vigorous activity, leisure time activities, sitting/reclining, walking, and sleeping”. How were low intensity activities classified? Was taking for granted that all leisure time activities were low intensity activities?

- “We assessed the dietary habit of the mother using a food frequency questionnaire within a month, prior to the baseline survey”. Does the food questionnaire used reflect the food habits of the study population? Please, briefly clarify it, as it seems you did.

- “Depending on their engagement in daily physical activity, mothers were categorized into the moderate intensity physical activity group and vigorous intensity physical activity group”. Were low activity women classified as moderate? Or there were no women who developed a low quantity of physical activity? Please, clarify.

- Is it possible to include information about how much activity or intensity was considered to classify women as "moderate" or "vigorous"?

RESULTS

- Please, the first time ANC is mentioned (Mothers’ obstetric history, behavioral characteristics, and anthropometric measurement section), clarify what this means (antenatal consultation?).

- I strongly suggest adding Body Mass Index (BMI) in the anthropometric measurement. Given that you have the height and the pre-pregnancy weigh, it is easy to calculate it, and it could add additional information that may be more representative of the reality than only the height or the weigh by itself.

- I also suggest adding information about BMI in Table 2 to see whether there were more LBW babies among women in any of the BMI categories (maybe underweight?). Even if the aim of the study is to analyze the possible correlation of the level of physical activity with birtweight, the study seems quite complete and provides extra information about socioeconomic, obstetric and behavioral characteristics, and food consumption; therefore BMI could add additional information, since it usually have a strong relationship with physical exercise and nutrition.

- Please, change name of Table 2 (it has been called Table 1 by mistake).

- What was the reason why there is so little women with the information about gestational weight gain? It seem one of the reasons is because little women reported their pre-gestational weight. Please, explain it.

- Table 3: please, modify the order in the enset, vegetables consumption, fruits, legums and sweets rows (4-6 times/week before 2-3 times/week).

- Please, revise and correct the third sentence of the section called “Physical activity level and incidence of low birthweight”, It is mentioned that “The incidence of LBW was significantly higher among mothers in the vigorous activity group 12 (21.6%) than moderate 12 (9.7%) activity group (p = 0.011)”. I believe you mean 24 women in the vigorous activity group; otherwise the numbers do not fit.

- Please, add the statistical significance of the next sentence: “The incidence of LBW was 25 (21.2%) and 4 (11.1%) in mothers who walked more than 60 minutes and below 30 minutes per day, respectively (, ADD with no statistical significance? or the P value)”.

- Please, correct the location of the percentages in the last sentence of the “Physical activity level and incidence of low birthweight” section. It can be misunderstanded: “With regard to the level of sedentary physical activity, 34 (15.7% REMOVE) (ADD 94.4%) and 26 (16.4% REMOVE) (ADD 72.2%) of the mothers who gave birth to LBW babies were those who sat or reclined for <165 minutes per day (HERE 15.7%) and slept ≥8 hours (HERE 16.4%), respectively”.

- Please, do not refer to sleep as sedentary physical activity, since sleeping is not a type of physical activity.

- Please, correct the sentence in page 16, line 268: “The risk of LBW was 2.6 times higher in mothers who were in their third trimester (ADD: AND PERFORMED SQUATTING) than those who did not perform squatting”.

6. PLOS authors have the option to publish the peer review history of their article (what does this mean?). If published, this will include your full peer review and any attached files.

Reviewer #1: Yes: Caroline Homer

Reviewer #2: Yes: Marina Vargas-Terrones

---

## [Author Response · Author response to Decision Letter 0]

24 Jun 2020

Response to Reviewers

Ref.: PONE-D-20-09704

Level of physical activity during the third trimester of pregnancy and its association with birthweight at term in South Ethiopia: A prospective cohort study 

PLOS ONE 

I, on behalf of all authors, thank editors and reviewers for their valuable and constructive comments to improve the excellence of this paper. We revised the manuscript based on the comments from the editor and the reviewers. Here is point by point response for each comments and questions raised by editors and reviewers.

Editors

SPECIFIC ACADEMIC EDITOR COMMENTS: Thank you for submitting your manuscript. Two expert reviewers handled your manuscript. Although interest was found in your study, there were major comments that arose during review. A number of these comments relate to the need for clarification and expansion of several vague points throughout the manuscript. Please note while forming your response, if your article is accepted, you may have the opportunity to make the peer review history publicly available. The record will include editor decision letters (with reviews) and your responses to reviewer comments. If eligible, we will contact you to opt in or out.

Response:

If the article is accepted, we agree that the peer review history to be publicly available.

Response:

Thank you for your comment. We revised the manuscript and we hope that the revised version meets all the journal requirements.

Response: 

The questionnaires (English and the local language – Amharic) are uploaded as supporting information.

Response:

We uploaded the minimal anonymized data-set as supporting information.

Please provide an amended statement that declares *all* the funding or sources of support (whether external or internal to your organization) received during this study, as detailed online in our guide for authors at http://journals.plos.org/plosone/s/submit-now. Please also include the statement “There was no additional external funding received for this study.” in your updated Funding Statement. Please include your amended Funding Statement within your cover letter. We will change the online submission form on your behalf.

Response:

Here is the updated funding statement;

“ML has received partial financial support from Addis Ababa University for data collection. All funders had no role in the study design, data collection and analysis, decision to publish, or preparation of the manuscript. There was no additional external funding received for this study.”

5. Please include a copy of Table 2 which you refer to in your text on page 11.

Response: Table 2 is available in page 12 of the revised manuscript.

Reviewer #1: 

Thank you for the opportunity to review this paper. The paper examines the relationship between physical activity and low birth weight in a cohort of women in Ethiopia. The topic is important as LBW is a significant contributor to newborn mortality and morbidity and had longer impacts on childhood growth and development. More detail is needed in a number of areas.

1. What gestation were women recruited? The third trimester is too broad and better clarity is needed. How was the gestation determined and how sure are the authors of the accuracy of the gestation?

Response: 

Thank you for your comment. We enrolled women in their 31 to 34 weeks of gestation. The gestational age was determined using the last menstrual period (LMP). We add a description in line 108-110 of the revised version. Since the participants are under ongoing pregnancy surveillance and the list was taken from it, the gestational age would not be affected to a large extent. Even if there is a possibility of mis-calculation of the GA, the effect size would not be affected due to non-differential for both exposed and unexposed groups.

2. The inclusion states that women with a preterm baby and multiple birth were excluded. How were multiple births determined? Does the preterm birth comment mean women with a history of previous preterm birth were excluded as whether they will have a preterm birth cannot be known at this point?

Response: 

Thank you for your comment. We excluded those with previous pre-term baby before enrollment. As we didn’t have access to ultrasound, we had planned to exclude preterm birth and multiple birth from the analysis (not from enrollment-as we couldn’t confirm it with ultrasound). However, there was no multiple birth. There was 1 who delivered pre-term in the current pregnancy and was excluded from the analysis. To make it more clear, we revised it in the study design and analysis section.

3. Clarity about when the physical activity was taken from is needed. Many women do slow down towards the end of pregnancy so their activity level in early pregnancy may be different to late pregnancy. These things are not statistic necessarily. Was this accounted for at all?

Response:

All PA measurements taken during enrollment (31-34 weeks of gestation). The questionnaire measured the level of physical activity of the women in the last 7 days prior to data collection time. We described it in line 122-131 of the revised version.

4. In the analysis section, a clearer explanation of how the variable for the multivariate analysis were selected is needed. What was the cutoff point? Which ones were initially included and then removed?

Response:

A bivariable analysis was carried out to determine the crude association of predictor variables with birthweight, and based on the results, variables with p-value <0.05 were selected for multivariable analysis. The variable ANC follow up was excluded from the multivariable model due to collinearity with iron folic acid supplementation. We described it in line 163-168 of the revised manuscript.

5. Parity does not seem to have had much attention. This is important as it is associated with a number of important issues including anemia. Why was it not included in the multivariate analysis?

Response:

We agree with your concern. Gravidity and parity are presumed to be risk factors of low birthweight. However, in our sample their confounding effect is minimal. The p-value of both gravidity and parity during bivariate analysis was high (0.74 and 0.498 respectively) as presented in table 2, page 12. This doesn’t mean they are not risk factors, but, their confounding effect in our sample is very minimal. Therefore, we did not included them in the multivariate analysis. Hence, their impact on the association between physical activity and low birthweight in our sample is minimal. 

6. Abortion is included which is important. Is this spontaneous abortion which would be better mentioned as a miscarriage or an induced abortion? The long term implications of these two are different and so need to be clarified.

Response:

It is spontaneous abortion or miscarriage. We revised it throughout the manuscript as miscarriage. 

7. BMI is a common measure in studies like this but I cannot see this included.

Response:

We used mid-upper arm circumference (MUAC) and it is the good surrogate during late pregnancy (third trimester). Several studies recommended MUAC as an indicator of nutritional status of mother during pregnancy, especially for in such context, where mothers hardly monitor their pre-pregnancy and post-pregnancy weight.

8. The food consumption analysis is comprehensive but not easy to interpret. There are also too many individual comparisons related to fruit and vegetables etc. consumed. Can these data be summarized into adequate diet versus not adequate diet?

Response:

Thank you for your comment. We agree that, it would be better to have a more comprehensive composite measure like ‘adequate/inadequate’. However, we couldn’t categorize it to adequate and inadequate as we didn’t measure the exact amount of food they took. We just assessed the relative frequency of intake for each food items. The weighed food diaries were not practically feasible in our context. We admit this limitation and add a description in line 344-346 of the revised version.

9. There are more than 30 individual comparisons which makes the chance of finding an error when one does not really exist high, especially given the sample size. Can the authors comment on the power of the study given the number of comparisons and the possibly of statistical error?

Response:

Thank you for your concern. We agree that type one error might be inflated. However, since our main hypothesis is to see the association between physical activity and low birthweight, other variables were assessed to see the confounding effect. In the meantime, we reported variables that remain significant during multivariate analysis. The study is powered to detect to effect of PA on LBW. The power for other predictors may not be sufficient. 

10. The multivariate analysis seems to be missing important issues – BMI in early pregnancy, parity, anemia? Were interactions examined? It is possible that there is an interaction between physical activity, standing for long hours and squatting. Given these are the three outcomes with significant results further examination needs to occur.

Response: 

We used MUAC instead of BMI, as we described in the previous response. Regarding the pre-pregnancy or early pregnancy BMI, we used self-reported weight to compute gestational weight gain. However, the pre-pregnancy weight is prone to error due to low educational level of study participants. In addition, only 25% of women reported their pre-pregnancy weight. Regarding parity, we agree that it is a known risk factor. However, in our sample it is comparable in both groups and the bivariate analysis also showed that as we mentioned in the previous response. Therefore, the confounding effect is minimal in this study. 

11. Some of the language needs attention. For example, scant rather than scanty (line 76), women instead of patients (line 99) and gave birth to rather than delivered (line 180).

Response: 

Thank you for your suggestion. We revised the whole manuscript for grammar and spelling. 

Reviewer #2: 

The manuscript reports a prospective cohort study addressing the association of the level of physical activity with birthweight in Ethiopia, where there is a high incidence of low birthweight, and women work and house conditions usually lead to high volumes and intensities of physical activity. The study is well designed, the methodology is well explained, and the paper is well written and with a good level of English. Every section contains the necessary information and they are well related to each other. Just a few comments and small corrections have been suggested:

OVERVIEW:

1. The study seems quite complete. However, even if physical activity is the main outcome, I believe the study covers much more than physical activity. The study provides additional information about how other factors may be associated with low birthweight, such as socioeconomic, obstetric and behavioral characteristics, food consumption and iron and folic acid supplementation. Therefore, I would extend the title of the paper to show that other factors have been studied, for instance: “Level of physical activity and other maternal characteristics during the third trimester of pregnancy and its association with birthweight at term in South Ethiopia: A prospective cohort study”. If the suggestion is followed, I would also mention this in the abstract.

Response: 

Thank you for your suggestion. We agree that other relevant determinants are also studied. We accept the suggestion and we mentioned it in the title and abstract.

METHODS

2. Please, explain when this study was carried out.

Response: 

Thank you for your comment. We added the information that the study was conducted from January to June 2017 in line 88-89 of the revised version.

3. Please, explain if the modified version of the global physical activity questionnaire was validated or published somewhere. Was the pilot test in Silte published?

Response: 

No, the pilot test is not published. The GPAQ is already internationally validated standard questionnaire. We customized it for the types of activities specific to the context of the study area. We performed an internal consistency test using the pilot data, and results showed acceptable level (Cronbach’s alpha = 0.79). We mentioned it in line 137-141 of the revised manuscript.

4. Does the questionnaire consider the intensity of the physical activity developed? Or was the intensity deduced according to the type of exercise? “The level of physical activity was classified as moderate activity, vigorous activity, leisure time activities, sitting/reclining, walking, and sleeping”. How were low intensity activities classified? Was taking for granted that all leisure time activities were low intensity activities?

Response: 

Thank you for your comment. The intensity is estimated according to the type of exercise. Regarding leisure time activities, it is editing problem. Leisure time activities are not part of the level of physical activities rather the domain or the type of activities. As you said leisure time or recreational activities could also be vigorous. We also divided leisure time into vigorous and moderate. For instance, running continuously for at least for 10 minutes is considered as vigorous despite it is recreational. We revised it in line 124-129 of the method section. For more information, we uploaded the study questionnaires as supporting information.

5. “We assessed the dietary habit of the mother using a food frequency questionnaire within a month, prior to the baseline survey”. Does the food questionnaire used reflect the food habits of the study population? Please, briefly clarify it, as it seems you did.

Response:

Thank you for your comment. We assessed the usual intake and frequency of food intake during pregnancy. In general, Food Frequency Questionnaires (FFQ) can be asked 1 month, 3 month, 6 month, 1 year or 5 year prior to the data collection time depending on the aim of the study. Usually chronic disease studies use one or five year dietary history in order to assess the long term impact of diet. However, in our case we were concerned on the dietary habit during pregnancy. Therefore, we used dietary history of 1 month prior to the data collection point. 1 month is preferable in a situation like us to minimize recall bias. We revised it in line 131-134 of the revised version.

6. “Depending on their engagement in daily physical activity, mothers were categorized into the moderate intensity physical activity group and vigorous intensity physical activity group”. Were low activity women classified as moderate? Or there were no women who developed a low quantity of physical activity? Please, clarify.

Response:

Thank you for your comment. There were few women who had low intensity physical activity and were categorized as moderate PA level. Since our aim was to assess the impact of vigorous physical activity on low birthweight, we considered those who do not have vigorous activity as one group (both moderate and low). We add more descriptions in line 128-130 of the revised version.

7. Is it possible to include information about how much activity or intensity was considered to classify women as "moderate" or "vigorous"?

Response:

We used the WHO GPAQ guideline. We ask them “Does your work involve vigorous-intensity activity that causes large increases in breathing or heart rate?” by showing them cards to specific activity types such as carrying, loading or stacking wood; chopping wood-splitting logs; manual grinding; drawing water from the well, river, etc. Similarly for leisure time physical activities. We then categorized those who do any type of vigorous activities in one group and those who do not in another group. The type of vigorous activities reported were predominantly work related activities.

RESULTS

8. Please, the first time ANC is mentioned (Mothers’ obstetric history, behavioral characteristics, and anthropometric measurement section), clarify what this means (antenatal consultation?).

Response:

Thank you for your comment. We write it in long form in its first appearance.

9. I strongly suggest adding Body Mass Index (BMI) in the anthropometric measurement. Given that you have the height and the pre-pregnancy weight, it is easy to calculate it, and it could add additional information that may be more representative of the reality than only the height or the weight by itself.

Response:

Thank you for your comment. We also agree that BMI could have additional information. However, only 25% of women reported their pre-pregnancy weight. On the other hand, using self-reported pre-pregnancy weight to calculate pre-pregnancy BMI is somehow prone to error. The main reason to ask pre-pregnancy weight was to estimate the gestational weight gain. Instead, We used mid-upper arm circumference (MUAC), and it is a good surrogate during late pregnancy (third trimester).

10. I also suggest adding information about BMI in Table 2 to see whether there were more LBW babies among women in any of the BMI categories (maybe underweight?). Even if the aim of the study is to analyze the possible correlation of the level of physical activity with birthweight, the study seems quite complete and provides extra information about socioeconomic, obstetric and behavioral characteristics, and food consumption; therefore BMI could add additional information, since it usually have a strong relationship with physical exercise and nutrition.

Response:

Thank you for your comment. We agree that including BMI might be additional information. If the pre-pregnancy weight and height was measured by the data collectors, it would have been reasonable to use it. However, due to the pre-pregnancy weight is self-reported, it is prone to error as the women in the study population has relatively low level of education. Only 25% of women reported their pre-pregnancy weight, which make using BMI less reliable. The self-reported weight is a simple rough estimate. Moreover, the height is measured during enrollment, using self-reported pre-pregnancy weight and height during 3rd trimester would have flawed results. Instead, we used MUAC measured by trained data collectors, which we think it is acceptable during pregnancy.

11. Please, change name of Table 2 (it has been called Table 1 by mistake).

Response:

Thank you for your comment. We revised it accordingly. 

12. What was the reason why there is so little women with the information about gestational weight gain? It seem one of the reasons is because little women reported their pre-gestational weight. Please, explain it.

Response:

Yes, due to low educational status of women in the study area, it is rare for them to measure and monitor their weight. That is the reason we do not use pre-pregnancy BMI. We described in line 348-349 of the revised version.

13. Table 3: please, modify the order in the enset, vegetables consumption, fruits, legums and sweets rows (4-6 times/week before 2-3 times/week).

Response:

Thank you for your comment. We revised it accordingly.

14. Please, revise and correct the third sentence of the section called “Physical activity level and incidence of low birthweight”, It is mentioned that “The incidence of LBW was significantly higher among mothers in the vigorous activity group 12 (21.6%) than moderate 12 (9.7%) activity group (p = 0.011)”. I believe you mean 24 women in the vigorous activity group; otherwise the numbers do not fit.

Response:

Thank you for your feedback. Yes, it is 24 (21.6%).

15. Please, add the statistical significance of the next sentence: “The incidence of LBW was 25 (21.2%) and 4 (11.1%) in mothers who walked more than 60 minutes and below 30 minutes per day, respectively (, ADD with no statistical significance? or the P value)”.

Response: Thank you for your comment. We added p-value though not statistically significant. 

16. Please, correct the location of the percentages in the last sentence of the “Physical activity level and incidence of low birthweight” section. It can be misunderstanded: “With regard to the level of sedentary physical activity, 34 (15.7% REMOVE) (ADD 94.4%) and 26 (16.4% REMOVE) (ADD 72.2%) of the mothers who gave birth to LBW babies were those who sat or reclined for <165 minutes per day (HERE 15.7%) and slept ≥8 hours (HERE 16.4%), respectively”.

Response: 

Thank you for your comment. We re-arranged the percentages.

17. Please, do not refer to sleep as sedentary physical activity, since sleeping is not a type of physical activity.

Response: 

Thank you for your comment. We separated ‘sleep’ from sedentary lifestyle throughout the manuscript.

18. Please, correct the sentence in page 16, line 268: “The risk of LBW was 2.6 times higher in mothers who were in their third trimester (ADD: AND PERFORMED SQUATTING) than those who did not perform squatting”.

 Response: 

Thank you for your comment. We revised it accordingly.

With regards,

Hamid Y. Hassen

---

## [Decision Letter · Decision Letter 1]

30 Jun 2020

Level of physical activity and other maternal characteristics during the third trimester of pregnancy and its association with birthweight at term in South Ethiopia: A prospective cohort study

PONE-D-20-09704R1

Dear Dr. Hassen,

We’re pleased to inform you that your manuscript has been judged scientifically suitable for publication and will be formally accepted for publication once it meets all outstanding technical requirements.

Kind regards,

Frank T. Spradley

Academic Editor

PLOS ONE

Reviewers' comments:

Reviewer's Responses to Questions

**Comments to the Author**

1. If the authors have adequately addressed your comments raised in a previous round of review and you feel that this manuscript is now acceptable for publication, you may indicate that here to bypass the “Comments to the Author” section, enter your conflict of interest statement in the “Confidential to Editor” section, and submit your "Accept" recommendation.

Reviewer #1: All comments have been addressed

2. Is the manuscript technically sound, and do the data support the conclusions?

Reviewer #1: Partly

3. Has the statistical analysis been performed appropriately and rigorously? 

Reviewer #1: I Don't Know

4. Have the authors made all data underlying the findings in their manuscript fully available?

Reviewer #1: Yes

5. Is the manuscript presented in an intelligible fashion and written in standard English?

Reviewer #1: Yes

6. Review Comments to the Author

Reviewer #1: The authors have addressed my comments mostly. I still have two problems with the paper though - the way diet is reported which is very hard to interpret and the very many comparisons mean that it is possible that differences were found by chance.

The other problem is the possible interaction between the some of the activities which I explained in my initial review. I still have concerns that the activities are not mutually exclusive.

The authors have addressed these but I am not sure their actual design makes it possible to completely address these issues.

7. PLOS authors have the option to publish the peer review history of their article (what does this mean?). If published, this will include your full peer review and any attached files.

Reviewer #1: **Yes: **Caroline Homer

---

## [Editor Report · Acceptance letter]

6 Jul 2020

PONE-D-20-09704R1 

Level of physical activity and other maternal characteristics during the third trimester of pregnancy and its association with birthweight at term in South Ethiopia: A prospective cohort study 

Dear Dr. Hassen:

I'm pleased to inform you that your manuscript has been deemed suitable for publication in PLOS ONE. Congratulations! Your manuscript is now with our production department. 

Kind regards, 

on behalf of

Dr. Frank T. Spradley 

Academic Editor

PLOS ONE